# Endometriosis and Cancer: Exploring the Role of Macrophages

**DOI:** 10.3390/ijms22105196

**Published:** 2021-05-14

**Authors:** Daria Artemova, Polina Vishnyakova, Elena Khashchenko, Andrey Elchaninov, Gennady Sukhikh, Timur Fatkhudinov

**Affiliations:** 1Scientific Research Institute of Human Morphology, 117418 Moscow, Russia; artiomova.darya@yandex.ru (D.A.); tfat@yandex.ru (T.F.); 2National Medical Research Center for Obstetrics, Gynecology and Perinatology Named after Academician V.I., Kulakov of Ministry of Healthcare of Russian Federation, 117997 Moscow, Russia; khashchenko_elena@mail.ru (E.K.); elchandrey@yandex.ru (A.E.); gt_sukhikh@bk.ru (G.S.); 3Department of Histology, Cytology and Embryology, Peoples’ Friendship University of Russia (RUDN University), 117997 Moscow, Russia; 4Histology Department, Pirogov Russian National Research Medical University, 117997 Moscow, Russia

**Keywords:** endometriosis, macrophages, monocytes, cancer, polarization

## Abstract

Endometriosis and cancer have much in common, notably their burgeoning of cells in hypoxic milieus, their invasiveness, and their capacity to trigger remodeling, vascularization, and innervation of other tissues. An important role in these processes is played by permissive microenvironments inhabited by a variety of stromal and immune cells, including macrophages. Remarkable phenotypical plasticity of macrophages makes them a promising therapeutic target; some key issues are the range of macrophage phenotypes characteristic of a particular pathology and the possible manners of its modulation. In both endometriosis and cancer, macrophages guard the lesions from immune surveillance while promoting pathological cell growth, invasion, and metastasis. This review article focuses on a comparative analysis of macrophage behaviors in endometriosis and cancer. We also highlight recent reports on the experimental modulation of macrophage phenotypes in preclinical models of endometriosis and cancer.

## 1. Macrophages (an Overview)

Macrophages are mononuclear cells of innate immunity, found ubiquitously throughout the body and indispensable for tissue homeostasis. Macrophages are classified by their origin from different progenitor cells over the course of individual development. Such classification emphasizes the distinction of tissue resident macrophages (derived from the yolk sac mesenchyme and embryonic liver), as opposed to bone marrow-derived monocytes which circulate with the blood and eventually die by apoptosis unless recruited to tissues [1]. Under the influence of local microenvironments, the recruited monocytes differentiate into macrophages and dendritic cells [2].

Macrophages implement several functions, including (1) the phagocytosis of pathogens, compromised cells, and debris; (2) antigen presentation; and (3) cytokine production. Macrophages have been implicated in many pathological processes, both inflammatory and proliferative [3].

Macrophages can either support or counteract inflammation, depending on the nature of activating signals. Proinflammatory macrophage activation is triggered by exposure to pathogen-associated molecular patterns (PAMPs). Recognition of PAMPs by macrophages makes them produce proinflammatory cytokines TNF-α, IL-1β, and IL-6. The resulting full-fledged inflammatory reaction facilitates additional recruiting of monocytes and other leukocytes from circulating blood; however, excessive inflammation is destructive and prevents healing. The negative effects of inflammation can be reduced by so-called alternative activation of macrophages towards anti-inflammatory phenotypes [4]. The distinction between proinflammatory and anti-inflammatory macrophage phenotypes is pivotal for understanding inflammation as a phenomenon. Importantly, macrophage polarization is reversible: the cells retain their plasticity, i.e., they can switch phenotypes depending on the current demands of the microenvironment. Pro- and anti-inflammatory phenotypes of macrophages became the basis of the so-called M1/M2 paradigm by analogy with the similar classification of T helpers. With regard to the existence of intermediate macrophage phenotypes (sometimes called hybrid phenotypes) [5], the M1/M2 paradigm has been repeatedly criticized as simplistic [6]. Specht et al. revealed the continuum of phenotypes in mature macrophage populations even in the absence of polarization inducers; the authors applied single-cell proteomic analysis to a random sample of macrophages from a single noninduced pure culture [7]. Furthermore, the continuous nature of macrophage plasticity manifests itself under both physiological and pathological conditions [8]. The question on switching between the pro- and anti-inflammatory states has been extensively described in a number of excellent reviews [8,9]. However, many authors still publish macrophage-related research with reference to the M1/M2 paradigm because this designation has become traditional and widespread [10,11,12,13,14].

The M1/M2 dichotomy is highly advantageous for experimental research as it conveniently reflects polar states of macrophage differentiation; nevertheless, the concept of polarity should be used with caution, especially for in vivo models. Here, we will use the terms “pro-” and “anti-inflammatory” to designate the most polar phenotypes observed in the reviewed studies.

The polarization of macrophages requires the activation of certain intracellular signaling cascades. Anti-inflammatory (M2) polarization is induced by cytokines IL-4 and IL-13, as well as IL-10 and IL-33, binding to their cognate receptors at the cell surface. Anti-inflammatory macrophages (M2-polarized) produce IL-12 at low levels and IL-10 and TGF-β at high levels; they actively scavenge pathogens and tissue debris by phagocytosis; and they promote healing and tissue remodeling while stimulating angiogenesis [3,15]. In addition to IL-4, IL-13, IL-10, and IL-33, the alternative activation of macrophages can be induced by other signals. To distinguish between various induction options, the alternatively activated macrophages have been subdivided into four subtypes: M2a, M2b, M2c, and M2d. In particular, M2a are induced by IL-4 and IL-13, while M2b are induced by immune complexes and LPS. It is noteworthy that M2b macrophages express certain markers of proinflammatory polarization, such as CD86, IL-1, IL-6, and TNF-α. M2c are induced by glucocorticoids and TGF-β, while M2d are induced by IL-6 and adenosines [16,17].

The polarization of macrophages is marked by the production of corresponding cytokines, chemokines, and receptors (Table 1) [15,16,17,18].

Thus, macrophages are versatile cells that can effectively react to various stimuli by polarization into a spectrum of phenotypes with the extremes designated M1 (proinflammatory) and M2 (anti-inflammatory). The major involvement of macrophages has been demonstrated for a number of pathological conditions (sepsis, cancers, metabolic syndrome, immune deficiencies, and autoimmune disorders), indicating their value as potential therapeutic targets. In the next section, we focus on endometriosis and characterize it for a subsequent comparison with cancer in the aspect of macrophage contributions to pathogenesis.

## 2. An Overview of Endometriosis

Endometriosis is an estrogen-dependent chronic inflammatory disease [19] manifested by the presence of endometrial-like tissue outside the endometrium [20,21]. The term was coined by Blair Bell in 1892; since then, more than 30 classifications of endometriosis have emerged. By its topography and severity, endometriosis can be classified into three clinical phenotypes: peritoneal surface lesions, endometrioid ovarian cysts, and deep-infiltrating endometriosis (DIE) [22]. The latter represents an especially severe form of the disease, with nodules located deeper than 5 mm beneath the peritoneal surface of the pelvic organs (rectum, urinary bladder, etc.) [23]. DIE is typically multifocal, and the nodules can be found at more distant extragenital locations, including the pleura, diaphragm, skin, eyeball, liver, and brain [22,24].

Endometriosis presents with chronic pelvic pains (associated with menstrual bleedings or acyclic) and dyspareunia; it may cause pelvic fibrosis (adhesions) and infertility [24,25]. Recent evidence suggests that 1 in 10 women of reproductive age lives with endometriosis, which amounts to estimated 176 million women globally [26,27]. Moreover, according to the Endometriosis Association, early onset is typical: about 50% of the patients experience their first symptoms under the age of 25 years, including 21% aged under 15 years and 17% aged 15–19 years [28].

The cellular sources of endometriotic lesions are diverse. The settlers are multipotent cells that most probably originate from the basal layer of endometrium, although alternative sites of their origin (mullerian ducts, bone marrow, and peritoneal mesothelium) cannot be excluded. The capacity of endometrial (endometrial-like) stem cells for cyclic focal growth at abnormal locations apparently results from a combination of hereditary and environmental factors, notably hyperestrogenism and pelvic hypoxia [29].

The mainstream paradigm of endometriosis is the theory of retrograde menstruation [19,22,30]. Sneaking into the pelvic cavity through the fallopian tubes, endometrial cells and tissue fragments may disseminate, adhere to mesothelium, and engraft, thus giving rise to endometrioid heterotopias; the effect is reinforced by subsequent ingrowth of blood vessels at the site of engraftment. Such a scenario implies that certain flaws in the local immune surveillance make it blind to the invasion/engraftment [19,22]. An important amendment to this theory has been made, accounting for the possibility that endometrial cells arrive in the pelvic cavity with neonatal uterine bleedings to be preserved in resting state until puberty, with subsequent deployment of endometrioid heterotopias under the incrementing estrogen levels.

However, distant heterotopias (located far away from the pelvic cavity at sites including the lungs, subarachnoidal space, skeletal muscles, kidneys, etc.) are more likely caused by the spread of endometrial or endometrial-like (endometrioid) tissue with lymph and blood flow [31]. Iatrogenic metastasis should be noted as well, including the emergence of endometriotic nodules in postoperative skin scars. Accordingly, endometriosis is viewed as a consequence of retrograde menstruation with the possible contributions of circulation-mediated spread and iatrogenic metastasis.

This concept, however, still fails to account for certain comorbidities (postmenopause, uterine agenesis) or endometriosis in males [32,33]. Such cases evoke further amendments to the theory. The concept of coelomic metaplasia links endometriosis to deviant (trans) differentiation of coelomic epithelium and its derivatives (which include ovarian stroma, mesothelium (peritoneal and pleural), renal tubules, lymphatic endothelium, etc.) under the influence of various hormonal and proinflammatory signals [24,34]. A related concept, known as the ‘embryonic rest theory’, assigns the leading role to cells of mullerian origin that settle and expand in peritoneal cavity while retaining the capacity of endometrioid differentiation under the action of estrogens [20]. Despite some obvious constraints of these concepts, they explain the sporadic incidence of endometriosis in males (affecting the bladder, testes, and prostate) [22].

It should be noted that transient endometriosis is physiological and apparently rather common; its transition into disease occurs under the influence of genetic, epigenetic, immunological, and hormonal factors. Compared to normal endometrial cells, the cells of endometrioid heterotopias show altered gene expression profiles [21,34]. These cells are highly proliferative; they show enhanced degrees of cell adhesion and invasiveness, reduced susceptibility to apoptosis, and a tendency to induce neoangio- and neurogenesis [21,28]. Similar properties, albeit at a more disastrous degree, are exhibited by cancer cells.

Endometriosis is an estrogen-dependent condition. An important role in its pathogenesis is assigned to altered proportions of estrogen and progesterone receptors against the background of local hyperestrogenism and progesterone resistance [24,28]. Endometrioid heterotopias reveal elevated levels of estrogen receptor β (ERβ) associated with reduced expression of ERα and progesterone receptor (PR), with the ultimate effect of progesterone resistance [24,28]. ERβ mediates the anti-inflammatory and proproliferative effects of estrogens on endometrioid heterotopias.

The structure of genetic predisposition to endometriosis is being investigated. Sapkota et al. (2017) confirmed the relevance of 9 of 11 loci previously implicated in endometriosis and associated five new loci (involved in sex hormone metabolism) with increased risks of endometriosis [35]. Local hyperestrogenism, which is a hallmark of endometriosis, can be associated with impaired expression of enzymes that metabolize estrogens (aromatases and 17β-hydroxysteroid dehydrogenases); the resulting local increase in estradiol levels stimulates cell growth within the lesion.

Macrophages are an integral component of endometriosis foci that is formed from three sources: endometrial macrophages, peritoneal macrophages, and macrophages differentiating from recruited blood monocytes. In the next section, we briefly characterize the main populations of macrophages involved in the development of endometriosis.

### 2.1. Endometrial Macrophages

Endometrium harbors a variety of immune cells subject to compositional changes during the menstrual cycle [36]. The dynamics of these changes indicate specific regulatory roles of different immune cells in the physiological collapse and recovery of endometrial tissues. The proliferation phase of the cycle is accompanied by the infiltration of endometrium with a mixed community of immune cells (macrophages, mast cells, T cells, and NK cells), which facilitate the recovery of the functional layer by promoting cell proliferation and morphogenesis [1].

As the functional layer matures, the proportion of macrophages among endometrial immune cells continuously increases, peaking by the end of secretory phase and desquamation [37,38], which suggests that menstruation has a strong inflammatory component [39,40]. During the secretory phase, resident endometrial macrophages (EMs) activate angiogenesis by producing vascular endothelial growth factor (VEGF). During menstruation, EMs apparently participate in triggering and control of the desquamation process; they also facilitate the subsequent recovery of the functional layer, including the restoration of glands [40].

The specific roles of EMs in menstruation have been associated with their phagocytic activity and abundant secretion of matrix metalloproteinases (MMPs), which digest the extracellular matrix, thus facilitating the detachment [1].

The cyclic changes in EM functionality are reflected by their protein expression profiles. EMs produce CD54, CD69, and CD71 during the proliferative phase [41]; VEGF during the secretory phase [42]; and MMP9, MMP12, and MMP14 during menstruation [43,44].

The heterogeneity of immune cells within endometrium implies concerted functionalities and separation of duties. The complex characterization of these cells and their interactions are necessary for understanding their functionalities under physiological conditions and in disease [38].

### 2.2. Peritoneal Macrophages

Peritoneal macrophages are the leading source of macrophage populations inhabiting the extragenital endometriosis foci. Peritoneal membranes are rich in macrophages which implement immune surveillance at peritoneal surfaces. Peritoneal macrophages (PMs) constitute over 50% of leukocytes found in human peritoneum [45]. PMs are subdivided into resident macrophages (rPMs) derived from embryonic hematopoietic sources and monocyte-derived (monocytic) macrophages (mPMs) of bone marrow origin [46,47]. rPMs are prominent phagocytes; human rPMs show a high expression of the immunoglobulin family complement receptor (CRIghi) and a low expression of C-C chemokine receptor type 2 (CCR2lo) [1,48]. By contrast, human mPMs have CRIgloCCR2hi phenotypes [48].

In mice, PMs are similarly subdivided into two distinct subpopulations of ’large’ and ’small’ cells. Large PMs, which predominate under physiological conditions, express F4/80 and CD11b at high levels (F4/80hiCD11bhi) and MHCII at low levels (MHCIIlo). Large PMs are considered a self-sustainable population descending from hematopoietic cells of the embryonic yolk sac. Small PMs are bone marrow derived and exhibit F4/80loMHCIIhi phenotypes, thus corresponding to the CRIgloCCR2hi PM population in humans [1]. Under inflammatory conditions, the ratio of PM subpopulations changes dramatically: large PMs decline, while small PMs increase in number [49,50]. This so-called “macrophage disappearance reaction” (MDR) is a hallmark of peritoneal inflammation [51]; the exact contributions of small PMs to its development and resolution remain underexplored [1].

Despite the correspondence of CRIghiCCR2lo/CRIgloCCR2hi and large/small subdivisions, human and murine PMs differ by the expression levels of certain transcription factors. For example, human CRIgloCCR2hi PMs express higher levels of GATA6 compared with small PMs in mice [48].

Macrophages express estrogen receptors, and macrophage functionalities are clearly estrogen dependent. Estrogens activate the proliferation of PMs, promote their anti-inflammatory polarization, and stimulate them to produce VEGF and MMPs [24,52]. All of these effects support the resolution of inflammation and subsequent healing [52].

On the other hand, prolonged experimental exposure of female rats or mice to estradiol results in an elevated production of proinflammatory markers by PMs, including IL1, IL6, TNFα, and iNOS [53,54]. The effect is ERα-dependent, as ESR1-/- knockout mice develop no proinflammatory response to estradiol under the same conditions [53,54]. Zhang et al. revealed a bipotential response to estrogens, which induce proinflammatory cytokine production by macrophages in low doses and inhibit this production when applied in higher doses [55].

Thus, the effects of estradiol on PMs are complex and depend on a huge variety of additional factors. The PM population reacts not only to the local inflammatory stimuli within the abdominal cavity but also to changes in systemic hormonal levels within the body. Throughout the course of this reaction, the synthesis of pro- and anti-inflammatory cytokines is activated, the proportion of resident macrophages in the macrophage population decreases, and the proportion of cells developing from hematopoietic stem cells of the red bone marrow increases.

### 2.3. Endometriosis-Associated Macrophages

The development of endometriosis foci is largely determined by microenvironments where endometrioid cells become allocated. An important cellular component of these microenvironments is constituted by immune system elements: lymphocytes and macrophages.

Endometriosis involves a local failure of T and NK cell-mediated immunity accompanied by an increased secretion volume of peritoneal fluid [24,47] and an increase in PM numbers [34].

The contributions of macrophages to endometriotic milieus are highlighted in many studies [21,56,57]. Histological analysis of endometriotic foci reveals elevated numbers of macrophages with distinct anti-inflammatory phenotypes [57,58]. It is currently thought that these anti-inflammatory macrophages facilitate endometriosis progression by stimulating proliferation of both epithelial and stromal cells within endometrioid heterotopias, while supporting their invasiveness and promoting angiogenesis, ultimately mediating the establishment of immunosuppressive microenvironments [1,21]. Macrophages have been implicated in endometriosis pains, as their interactions with nerve fibers favor the maintenance of inflammatory reaction and focal nociceptive hypersensitivity [25,59]. Macrophages tend to colonize endometriotic lesions, which release various specific chemoattractants (e.g., CSF1, CCL2, and MCP1) [25,59]. Colonization of the lesions by macrophages is accompanied by the repolarization of the latter towards the anti-inflammatory phenotype. Within the lesions, anti-inflammatory macrophages produce neuroactive peptides and growth factors (e.g., Sema3A, NGF, and VEGF), thus promoting local angio- and neurogenesis. For their part, nerve fibers within the lesions regulate the activities of macrophages; the reciprocal interactions between macrophages and nerve fibers aggravate the inflammation and nociception. Endometriosis-associated macrophages (EAMs) express elevated levels of insulin-like growth factor 1 (IGF1) and elevated concentrations of IGF1 in the peritoneal fluid of patients correlated with the intensity of endometriosis pains. IGF1 has been shown to facilitate neurogenesis and increase the sensitivity of nerve fibers [59]. In response to estradiol, nerve fibers secrete high amounts of CCL2 and CSF1, which attract macrophages; for their part, macrophages produce factors that facilitate nerve ingrowth [25,60].

Proinflammatory cytokines released by macrophages (e.g., IL1β, MCP1, and TNFα) activate the nociceptive TRPV1 channels on peripheral nerve endings. Nerve growth factor (NGF), which is also released by macrophages, facilitates the sprouting of nociceptors [44]. The interaction of the monocyte chemoattractant protein 1 (MCP1) with its receptor facilitates the sensitization of nerve fibers by reducing the excitation threshold via a CCR/Gβγ-dependent mechanism [25,61]. Persistent stimulation of peripheral nerve endings promotes the abundant release of inflammatory neuropeptides, notably substance P (SP) and the calcitonin gene related peptide (CGRP). The release of SP from nerve fibers activates its cognate receptor on macrophages (neurokinin 1 receptor, NK1R), thereby facilitating the secretion of proinflammatory factors (CXCL8, IL1β). At the same time, the activity of SP and CGRP may shift macrophage polarization towards the anti-inflammatory phenotype. Both the macrophage-mediated and neurogenic components of inflammation are estrogen dependent, which aggravates the peripheral nociceptive sensitization in endometriosis.

Despite the distinct anti-inflammatory properties of EAMs, their phenotypes remain understudied. Surface expression of CD163 and CD206 by EAMs is evident [57]; at the same time, EAMs express MMP17 [62] and plausibly also a combination of CD163 and iNOs [63]. Further studies of EAM properties are highly relevant, as an understanding of their pathogenic roles in endometriosis may open new therapeutic possibilities.

Anatomical localization of endometriotic lesions and certain features of their histogenesis suggest that macrophages involved in the pathogenic process originate from diverse sources. EAMs constitute a mixed population comprising resident macrophages from endometrium and peritoneum, as well as macrophages of bone marrow (monocytic) origin [1]. Studies on mice have demonstrated that endometrioid foci comprise a mixture of EMs with infiltrating monocytes and PMs [58,64,65]. Despite the definite involvement of PMs in the pathogenesis, their exact contributions remain uncertain, as does the role of infiltration with bone marrow-derived macrophages. The infiltrating monocytic macrophages initially exhibit proinflammatory phenotypes but subsequently develop elevated expression levels of arginase 1 and CD204, which correspond to alternative, tissue remodeling-associated phenotypes [66].

Endometriosis has been associated with characteristic alterations in the composition and behaviors of macrophages within eutopic endometrium of the uterus. For instance, CD163 + EMs decrease in number [67,68], while the proportion of CCL2 + EMs increases, indicating monocyte immigration [69]. Increased expression of MMP9 colocalizing with CD68 + cells may correspond to the enhanced detachment capacity and mobility of eutopic endometrium in endometriosis [70]. At the same time, the direct involvement of eutopic EMs in endometriosis is questionable.

The participation of PMs in endometriosis progression is more evident than the roles of other local macrophage populations. Profound alterations of PM functionality in patients with endometriosis have been reported [1,24]. Several studies demonstrate that the activation is mediated by the c-Jun/c-Fos/AP-1 pathway and provides no unidirectional PM polarization, which is consistent with the results obtained in animal models [56,71]. Other studies relate endometriosis to the anti-inflammatory polarization of PMs [57]; possible triggers of such polarization include IL17a produced by ectopic endometrioid cells [72,73]. Although the correlation of elevated levels of anti-inflammatory polarization markers (CD206, CCL7) expressed by PMs in the presence of IL17a with enhanced proliferation and vascularization is elusive [74], IL17a can stimulate angiogenesis [72], which is consistent with the reports on the enhanced production of VEGF by PMs as an important factor of growth and expansion of endometriotic foci [28,42]. Moreover, PMs of the patients may show certain signs of dysfunction, including impaired phagocytic activity due to insufficient expression of MMP9 [75]. Importantly, in endometriosis, PMs reveal immunosuppressive activity against NK cells [76].

Apart from the altered activities of PMs, endometriosis is marked with increased PM numbers [30,47]. Despite the well-established CRIghi/CRIglo distinction of human PMs, which is both phenotypical and functional, behaviors of these subpopulations in endometriosis remain unspecified [1]. The elevated expression of both pro- and anti-inflammatory cytokines by endometriosis-associated PMs [24,56] indicate that the ambivalent PM polarization and the lack of ‘macrophage disappearance reaction’ (MDR) are hallmarks of endometriosis. It has been demonstrated that inflammation proceeding at high levels of IL4 spares resident PMs [46]. High levels of IL4 observed in the peritoneal fluid of the patients [77] may promote the accumulation of resident PMs, thus supporting the invasion and proliferation of endometrial cells.

Thus, the distinguishing feature of extragenital endometriosis, as compared with other abdominal inflammatory processes, is the primary activation of resident PMs. Resident macrophages are known to be more sensitive to MCSF, IL10, and IL4 than bone marrow-derived macrophages [78,79,80], which can be used for selective targeting of resident PMs in endometriosis.

## 3. Tumor-Associated Macrophages

Tumor microenvironments comprise blood and lymph vessels, immune cells, and fibrous stroma [81,82]. The progression of a tumor is largely defined by its content of lymphocytes and macrophages with characteristic immunosuppressive profiles (high content of regulatory T cells and CD206+ macrophages, low content of cytotoxic T cells, high levels of TGFβ and IL10 production, and low levels of IL1β and IFNγ production).

Tumor-associated macrophages (TAMs) play a fundamental role in tumorigenesis. TAMs participate in all phases of the disease and associated processes, including oncogenic transformation, tumor cell proliferation, angiogenesis, invasion, and metastasis [83]. TAM behaviors are highly variable: they are capable of implementing both pro- and anti-inflammatory functionalities. In certain types of cancer, TAM infiltration may account for over 50% of the tumor mass [81,84]. Upon their recruitment to a nascent tumor, blood monocytes initially differentiate into proinflammatory TAMs switching their polarization towards anti-inflammatory phenotype(s) as the tumor matures [81,85]. Anti-inflammatory TAMs are generally considered protumorigenic because they suppress the inflammation and facilitate the extracellular matrix remodeling, thus promoting tumor growth [81,86]. Anti-inflammatory TAMs help to destroy basement membranes, promote angiogenesis, and suppress immune surveillance; they stimulate proliferation of tumor cells and support metastasis [4]. A number of excellent reviews [87,88,89,90] provide detailed analysis of TAM plasticity and its pathological consequences; we therefore restrict ourselves to a brief outline.

Endometriosis is frequently compared with estrogen-dependent cancers, as both conditions present with invasive growth, relapses after surgery, and metastasis [21,34]. Akin to endometriosis, tumor growth may depend on estrogens which exert mitogenic action and facilitate the expression of antiapoptotic proteins [91]. In both endometriosis and cancer, invasion involves MMP family proteins; for instance, MMP9 participates in both the implantation of ectopic endometrium in endometriosis and the invasion of tumor cells and metastasis in cancer [92,93]. Endometriosis has been associated with decreased expression of cyclin-dependent kinase inhibitor 1B (p27Kip1) in epithelial and stromal cells of endometrium [94]. p27Kip1 is a key effector of the G1/S cell cycle checkpoint, and its decreased expression promotes uncontrolled proliferation of endometrial cells. Decreased expression of p27Kip1 observed in certain cancers has been associated with poor prognosis [94]. Another common feature of endometriosis and cancer is suppressed apoptosis; accordingly, the inhibited expression of proapoptotic proteins and increased expression of antiapoptotic Bcl-2 family proteins are typical for both [93,95].

Similar to cancers, endometriosis has a strong genetic component, which refers to both the inherited predisposition and accumulation/selection of pathogenic somatic variants. Similar to tumors, endometrioid heterotopias tend to reprogram their microenvironments to support their growth (Table 2) [21,93].

## 4. Macrophages and Anticancer Therapy

With regard to their role in tumor progression, as well as their possible antitumor activity, macrophages are often considered as possible targets for antitumor therapy. For their part, proinflammatory macrophages exert antioncogenic properties: they are capable of killing tumor cells upon contact [102]. In addition, proinflammatory macrophages release cytokines that act as a ‘call for reinforcement’ by attracting additional components of the immune system to the tumor [4]. This property of proinflammatory macrophages is employed in a number of therapeutic approaches. At present, the most recognized immunity-based anticancer strategy is the use of cytokines as attractants for monocytic macrophages and simultaneously their proinflammatory polarization inducers [104]. A prominent example is FDA-approved IFNγ [84]; however, the odds of targeted delivery of IFNγ to solid tumors, as well as the unfeasibility of maintaining high local concentrations of the drug for a long time, complicates the widespread move of this approach to the clinic. Possible alternatives are based on macrophage reprogramming. The idea is to switch TAM phenotypes from anti- to proinflammatory with the use of biologically active mediators (inducers). The list of candidate inducers includes Toll-like receptor (TLR) agonists, monoclonal antibodies to proinflammatory polarization inhibitors, and inhibitors of anti-inflammatory polarization (e.g., metformin and histidine-rich glycoprotein).

TLR agonists constitute a promising group of anticancer drug candidates. Imiquimod binds TLR7; the binding leads to the nuclear translocation of NFκB in macrophages and results in the abundant production of proinflammatory mediators, notably TNFα, IL6, IL12, and CCL2 [105]. As an adjuvant, imiquimod shows significant synergistic effects with radiotherapy.

CpG oligodeoxynucleotide is another synthetic TLR agonist; its binding to TLR9 causes alterations in carbohydrate and fatty acid metabolism of macrophages, boosting their antitumor activity, notably the elimination of tumor cells, including CD47+ (“don’t eat me”-positive) cells by phagocytosis [106].

Proinflammatory macrophage behaviors can be induced by antibodies to CD40, a costimulatory protein expressed on antigen-presenting cells. CD40-specific antibodies stimulate the antitumor activities of macrophages, including the enhanced secretion of NO and TNFα [107]. CD40 represents a prospective therapeutic target for breast cancer, colon cancer, and melanoma; the mechanism involves the reprogramming of immunosuppressive anti-inflammatory macrophages towards proinflammatory phenotype(s) and the resulting enhancement of immune response within tumor foci. Similar results have been obtained for the surface marker CD24 hyperexpressed in ovarian cancer, whose macrophageal receptor Siglec-10 inhibits recognition of the tumor by the immune system. As demonstrated by Barkal et al., the use of antibodies to CD24 and Siglec-10 significantly enhances phagocytosis-mediated elimination for all CD24-expressing human tumors. In a similar way, Siglec-10 knockout (which also eliminates the CD24–Siglec-10 interaction) leads to a macrophage-dependent decrease in tumor growth and increased survival rates [108].

Macrophages express receptors to estrogen—a strong anti-inflammatory agent. Anti-inflammatory polarization of macrophages under the action of estrogens has been demonstrated [24,52]. Tamoxifen, an estrogen receptor antagonist which facilitates proinflammatory macrophage polarization, appears in various anticancer regimens.

Macrophage migration inhibitory factor (MIF) represents a promising target for cancer immunotherapy. MIF is known to facilitate progression and metastasis for a wide spectrum of malignant neoplasms; its elevated concentrations in various solid tumors have been associated with adverse prognosis. The use of MIF-specific short hairpin RNA was shown to suppress tumor growth [109].

STAT3 modulation is an attractive strategy for the induction of proinflammatory phenotypes in macrophages. A low-molecular STAT3 inhibitor WP1066 reverses immune tolerance in malignant glioma patients and selectively induces the expression of costimulatory proteins CD80, CD86, and IL12 on peripheral macrophages [110,111].

Thus, several promising molecular targets have been identified in macrophages, the modification of which may activate their antitumor properties. However, there are many difficulties along this path: the heterogeneity of tumors, the pronounced plasticity of macrophages, and the fact that these cells are difficult to be modified genetically. The evaluation of prospective clinical utility for these markers would require additional studies using in vivo models.

## 5. Macrophages and Anti-Endometriosis Therapy

Given the considerable generic similarity between tumorigenesis and endometriosis, the concept of using macrophages as the basis for anticancer therapy may have some useful inferences for the treatment of endometriosis. Murine models provide conflicting suggestions on the possibility of using macrophages to treat endometriosis. Some studies indicate increasing numbers of small (monocytic) PMs and decreasing numbers of large (resident) PMs in experimental endometriosis in mice, with both populations comprising a mixed content of pro- and anti-inflammatory cells [71]. Other authors observe no decrease in numbers of large (resident) PMs in murine models [59].

A recent study with the use of a syngeneic mouse model demonstrated the presence of both resident EMs of the donor and infiltrating macrophages of the recipient within ectopic foci [58]. The infiltrating subpopulation apparently comprises both resident PMs and bone marrow-derived monocytic macrophages; their contributions to the disease require further specification [1].

It has been widely admitted that endometriosis is an estrogen-dependent condition [19,24]. The influence of hyperestrogenic milieus on macrophage behaviors is of great importance for the development of anti-endometriosis therapies. The excess of estrogens affects the functionalities of endothelial and endometrial stromal cells. In particular, estrogens promote the expression of angiogenic factors, notably VEGF and TGFβ [112]; the latter has been associated with anti-inflammatory macrophage polarization [24,55]. Moreover, estradiol has been implicated in TGFβ-independent anti-inflammatory macrophage polarization [112]. Increased production of anti-inflammatory macrophage polarization markers and the predominance of anti-inflammatory macrophages within endometriotic foci have been demonstrated [21].

Considering the available experimental evidence, the administration of macrophages with distinct proinflammatory phenotypes holds certain promises for anti-endometriosis therapy. In the experiments with the transplantation of polarized macrophages to experimentally induced endometriotic foci, anti-inflammatory macrophages facilitated the expansion, whereas proinflammatory macrophages not only inhibited the expansion of ectopic endometrium but also promoted its disruption [21]. However, the value of such approaches is limited by the high plasticity of macrophages. Alterations in macrophage phenotypes under the influence of endometriotic microenvironments may reduce the positive effects of transplantation.

Another therapeutic strategy is the specific ablation of macrophages that support the development and expansion of endometriotic foci. Such attempts have been made in murine models. The elimination of macrophages by using liposomes loaded with bisphosphonates (e.g., clodronate) allowed significant reduction of endometriotic foci [57,58,59]. Some studies indicate the efficiency of the selective elimination of particular macrophage subpopulations. For instance, the elimination of infiltrating Tie2+ macrophages within a lesion promoted its regression [64]; similar results were obtained with the selective elimination of VEGF receptor 1-positive (VEGFR1+) macrophages [65]. A significant positive effect was also achieved with the ablation of PMs [113].

Another therapeutic possibility includes the inhibition of EAM proliferation with the use of CSF1R inhibitors [114] and the suppression of macrophage recruitment to endometriotic foci with the use of CCR2 blockers [115].

Thus, several therapeutic macrophage-targeting strategies for the treatment of extragenital endometriosis have been proposed so far. The evaluation of each strategy would require more experimental studies using in vivo models. Since resident macrophages are key regulators of tissue homeostasis, arbitrary control over their polarization and numbers may interfere with the functioning of different organs. In this regard, in addition to the experimental therapy effectiveness, the side effects should be also addressed comprehensively.

## 6. Conclusions

Noninvasive diagnostics and appropriate personalized immune cell therapies for endometriosis are highly relevant. Endometriosis is benign but resembles cancer in certain aspects of its onset, progression, and impact on cellular neighborhoods. New insights in the field of cancer may therefore facilitate the development of new therapies for endometriosis.

Macrophages represent an integral component of microenvironments within the foci of endometriosis, contributing to vascularization and disease progression. The important role of macrophages lies in the focus of several strategies proposed to treat endometriosis. These strategies include the activation (polarization) of macrophages towards phenotypes with anti-endometriotic activity, complete elimination of macrophages, or prevention of their migration to the foci. The development of macrophage-targeted anti-endometriosis therapies requires taking into account that this pathology is hormone dependent and that consistently altered levels of sex hormones can affect the state of the macrophage system.

Considering the described pathogenetic features of endometriosis development and progression, we believe that the most promising therapeutic strategy is the combined decrease in macrophages’ anti-inflammatory activation and local hyperestrogenic status. Moreover, cell therapy could be considered both in an independent form and as an addition to the hormone therapy already used. Due to the mutual influence of the estrogens and anti-inflammatory activated macrophages in the pathogenesis of endometriosis, strategies of directed modification of macrophage activity could be a useful tool in overcoming progesterone resistance and altering steroidogenesis in endometrioid heterotopias. Furthermore, immunocellular therapy would be of undoubted importance in increasing the effectiveness of surgical treatment with the possibility of targeted administration of directly polarized macrophages to the foci of endometriosis—for example, intraperitoneally.

Therapeutic macrophage reprogramming, which includes in situ and ex vivo options, was originally proposed for cancer. The similarity between endometriosis and cancer suggests the potential utility of such approaches for endometriosis management. The prospects include the facilitated demolition of endometrioid heterotopias by ex vivo re-programmed macrophages. In situ monitoring of macrophage behaviors may provide valuable predictors for anti-endometriosis therapy optimization. Despite the success in the controlled polarization of endometriosis-associated macrophages in mouse models, the available evidence on the specific roles of macrophages in this disease remains limited.

## Figures and Tables

**Table 1 ijms-22-05196-t001:** The spectra of cytokines, chemokines, and receptors expressed by macrophages with pro- and anti-inflammatory phenotypes [15,16,17,18].

Activating Signals	Markers Expressed by Macrophages
**Proinflammatory phenotype**	
IFN-Ɣ/ LPS/TNF-α or their combination	FcƔ-R I, II, III (CD16, CD32, CD64), TLR2, TLR4, CD80, CD86, TNF-α, IL-1α, IL-1β, IL-6, IL-12β, type I IFN, CXCL1, CXCL2, CXCL3, CXCL5, CXCL8, CXCL9, CXCL10, CXCL11, CCL2, CCL3, CCL4, CCL5, CCR7, CCR11, CCR17, CCR22, iNOS, ROI, IL-23A, HIF-1α, IRF1, Myeloid differentiation primary response gene (MyD88), STAT1, TRAIL, cyclooxygenase-2 (COX-2), NO
**Anti-inflammatory phenotype**	
IL-10	Scavenger receptor A (CD204), CD14, CCL16, CCR2
IL-4, IL-13	СD163, Fcε-RII (CD23), IL10, Decoy IL-1 R type II (a trapping receptor for IL-1), CCL17, CCL22, CCL24, CXCR1, CXCR 2
IL-4, IL-13, IL-10 IL-33	СD163, CD206, IL-1ra (IL-1 receptor antagonist), CCL18, arginase-1
Immune complexes, LPS	CD86, MHC II, IL-1, IL-6, IL-10, TNF-α
Glucocorticoids, TGF-β	CD163, CD206, IL-10, TGF-β, Mer tyrosine kinase (MERTK), Extracellular matrix (ECM)
IL-6, adenosines	VEGF-A, IL-10, IL-12, TNF-α, TGF-β

**Table 2 ijms-22-05196-t002:** Summary table showing the generic similarity between endometriosis progression and tumor growth with specification of the roles of macrophages.

Aspects of Pathogenesis	Manifestation in Tumorigenesis and Endometriosis Progression
Growth	Uncontrolled cell growth, invasion to surrounding tissues [34] A tendency to expand and promote neoangiogenesis and neurogenesis [28]
Architecture	Non-encapsulated lesions [34] The lesions are infiltrated with immune cells, vascularized, innervated, and have well-defined stroma [1,21,82]
Cell death	Dysregulation of the normal programmed cell death, suppression of apoptosis [93,95,96]
Inflammation and immunity	Sustained local inflammatory response [47,97]
Сellular metabolism	A shift in energy metabolism favoring glycolysis over mitochondrial respiration [98,99]
Attitude to hypoxia	Initial stages of tumorigenesis proceed under hypoxic conditions which promote HIF1α-dependent cell proliferation and angiogenesis [100]. Similarly, in nascent endometriotic foci, transient hypoxia promotes expression of VEGF via HIF1α activation [101]
The role of macrophages and macrophage-mediated influence of estrogens	Macrophages support either elimination or expansion (growth, spread) of the lesions [21,38,66,102]
	In nascent malignant tumors and endometriotic foci, macrophages acquire proinflammatory phenotype(s), whereas in mature lesions they exhibit anti-inflammatory phenotype(s) [38,66] Macrophages with anti-inflammatory phenotype(s) contribute to cancer and endometriosis progression [52,103]
	Estrogens support anti-inflammatory polarization of macrophages, thus facilitating growth of endometrioid heterotopias; the same effect of estrogens has been observed for tumors [52,103]

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
