# Peer review of "Endometriosis and Cancer: Exploring the Role of Macrophages"

_ijms, 2021, doi:10.3390/ijms22105196_

Round 1
Reviewer 1 Report
The review paper provides an overview of macrophage behaviors in endometriosis and cancer as well as a report on experimental modulation of macrophage phenotypes in preclinical models of endometriosis and cancer. However, the review fails to delineate current gaps in knowledge or provide commentary on current studies and future directions.
Comments:
Describing macrophages solely being M1/M2 phenotype is an outdated and overly simplistic classification. Authors need not get into the details of macrophage phenotypes and nomeclature but should at least mention more recent studies highlighting activation and inflammatory-to-anti-inflammatory states as a spectrum rather than a binary characterization.
Table 1 requires more references.
In my opinion, a review manuscript requires more than 1 reference for many of the points made throughout the manuscript.
Paragraphs end and start without forming complete thought.
Table 2 is difficult to read, but I imagine can be improved by reformatting.
Conclusion could be expanded upon to bring the review together into a story. Review provides a lot of information but could use more commentary on future direction of studies relating to macrophage role in endometriosis and cancer and how related studies can be applied to potential therapies.
Author Response
The overview paper provides an overview of macrophage behavior in endometriosis and cancer, as well as a report on experimental modulation of macrophage phenotypes in preclinical models of endometriosis and cancer. However, the review does not reflect existing knowledge gaps or comment on current research and future directions.
Dear reviewer, thank you for rating our article. We have tried to improve it according to your recommendations.
Comments:
Describing macrophages solely as the M1 / M2 phenotype is an outdated and oversimplified classification. The authors do not need to go into details of the phenotypes and nomenclature of macrophages, but at least mention should be made of more recent studies in which activation and inflammatory and anti-inflammatory conditions are considered as a spectrum rather than a binary characteristic.
Response: Thanks for this post. You have raised an important question. Indeed, the M1 / M2 paradigm has been overhauled over the past decade, and more and more criticism is causing it to be mentioned in articles. Despite this, many authors publishing works related to macrophages still refer to the M1 / M2 paradigm even in recently published works (Boucher et al., 2021; Moradi-Chaleshtori et al., 2021; Li et al., 2020 ; Yang et al., 2020; Palmieri et al., 2020).
As per your recommendation, we have relaxed the strict M1 and M2 naming of the M1 / M2 paradigm and added text regarding the macrophage phenotype continuum (lines 49–62, lines 65–67).
Boucher A., Klopfenstein N., Hallas V.M., Skibbe J., Appert A., Jang S.Kh., Pulacanti K., Rao S., Cowden Dahl K.D., Dahl R. MicroRNA cluster miR- 23a ∼ 27a ∼ 24-2 promotes inflammatory polarization of macrophages. J Immunol. 2021, February 1; 206 (3): 540-553. DOI: 10.4049 / jimmunol. 1901277.
Li L, Lv G, Wang B, Kuang L. 2020. The XIST / miR - 376c - 5p / OPN axis modulates the effect of proinflammatory macrophages M1 on chondrocyte apoptosis in osteoarthritis. Journal of Cellular Physiology 235: 281–293. DOI: 10.1002 / jcp.28968.
Yang SJ, Chen YY, Hsu CH, Hsu CW, Chang CY, Chang JR, Dou HY. 2020. Activation of M1 macrophages in response to recombinant anti-tuberculosis vaccines with increased antimycobacterial activity. Frontiers in Immunology 11: 1298. DOI: 10.3389 / fimmu.2020.01298.
Moradi-Chaleshtori M., Bandehpur M., Judge S., Mohammadi-Eganeh S., Hashemi S.M. In vitro and in vivo evaluation of the antitumor effect of the M1 phenotype induction in macrophages by exosomes containing miR-130 and miR-33. Cancer Immunol Immunother. 2021 May; 70 (5): 1323-1339. DOI: 10.1007 / s00262-020-02762-x.
Palmieri EM, Gonzalez-Cotto M, Baseler WA, Davies LC, Ghesquière B, Maio N, Rice CM, Rouault TA, Cassel T., Higashi RM, Lane AN, Fan TWM, Wink DA, McVicar DW. 2020. Nitric oxide controls metabolic changes in M1 macrophages by acting on aconitase 2 and pyruvate dehydrogenase. Connection with nature 11: 1-17. DOI: 10.1038 / s41467-020-14433-7.
Table 1 requires additional references.
Response: Thanks for this comment. In Table 1, we have added additional information on the phenotypes of macrophages and increased the number of references.
In my opinion, the review manuscript requires more than one reference for many of the points raised in the manuscript.
Response: Thanks for this post. We have added more links to support the thoughts expressed in the Review (marked in yellow).
The paragraphs end and begin without forming a coherent thought.
Response: Thanks for the note. On your recommendation, we have added an introductory part and a conclusion to the paragraphs related to the non-clinical sections of the manuscript. (lines 186 - 190, 219 - 220, 256 - 261, 263 - 266, 355 - 359, 400 - 401, 453 - 458, 460 - 462, 504 - 510, 537 - 557).
Table 2 is difficult to read, but I believe it can be improved by reformatting it.
Response: Thanks for your valuable comment. We have modified Table 2 to make it easier to understand.
The conclusion can be extended to integrate the overview into a story. The review contains a lot of information, but additional comments could be used about the future direction of research regarding the role of macrophages in endometriosis and cancer, and how related research might be applied to potential therapies.
Response: Thank you, we have expanded the Conclusion section and hope that it seems like a logical conclusion to the review as it stands.
В обзорном документе представлен обзор поведения макрофагов при эндометриозе и раке, а также отчет об экспериментальной модуляции фенотипов макрофагов в доклинических моделях эндометриоза и рака. Однако в обзоре не отражены существующие пробелы в знаниях или не даны комментарии к текущим исследованиям и будущим направлениям.
Уважаемый рецензент, спасибо, что оценили нашу статью. Мы постарались улучшить его в соответствии с вашими рекомендациями.
Комментарии:
Описание макрофагов исключительно как фенотипа M1 / M2 - устаревшая и чрезмерно упрощенная классификация. Авторам не нужно вдаваться в подробности фенотипов и номеклатуры макрофагов, но следует хотя бы упомянуть более поздние исследования, в которых активация и воспалительные и противовоспалительные состояния рассматриваются как спектр, а не бинарная характеристика.
Спасибо за эту заметку. Вы подняли важный вопрос. Действительно, парадигма M1 / M2 была пересмотрена за последнее десятилетие, и все больше и больше критики вызывают ее упоминания в статьях. Несмотря на это, многие авторы, публикующие работы, касающиеся макрофагов, по-прежнему ссылаются на парадигму M1 / M2 даже в недавно опубликованных работах (Boucher et al., 2021; Moradi-Chaleshtori et al., 2021; Li et al., 2020; Yang et al. al., 2020; Palmieri et al., 2020).
В соответствии с вашей рекомендацией мы смягчили строгое обозначение M1 и M2 парадигмы M1 / M2 и добавили текст, касающийся континуума фенотипа макрофагов (строки 49 - 62, строки 65 - 67) .
Баучер А., Клопфенштейн Н., Халлас В.М., Скиббе Дж., Апперт А., Джанг С.Х., Пулаканти К., Рао С., Кауден Даль К.Д., Даль Р. Кластер микроРНК miR-23a ∼ 27a ∼ 24-2 способствует воспалительной поляризации макрофагов. J Immunol. 2021, 1 февраля; 206 (3): 540-553. DOI: 10.4049 / jimmunol.1901277.
Li L, Lv G, Wang B, Kuang L. 2020. Ось XIST / miR - 376c - 5p / OPN модулирует влияние провоспалительных макрофагов M1 на апоптоз хондроцитов при остеоартрите. Журнал клеточной физиологии 235: 281–293. DOI: 10.1002 / jcp.28968.
Ян SJ, Chen YY, Hsu CH, Hsu CW, Chang CY, Chang JR, Dou HY. 2020. Активация макрофагов M1 в ответ на рекомбинантные противотуберкулезные вакцины с повышенной антимикобактериальной активностью. Границы в иммунологии 11: 1298. DOI: 10.3389 / fimmu.2020.01298.
Моради-Чалештори М., Бандехпур М., Суди С., Мохаммади-Еганех С., Хашеми С.М. Оценка in vitro и in vivo противоопухолевого эффекта индукции фенотипа M1 в макрофагах экзосомами, содержащими miR-130 и miR-33. Cancer Immunol Immunother. 2021 Май; 70 (5): 1323-1339. DOI: 10.1007 / s00262-020-02762-х.
Palmieri EM, Gonzalez-Cotto M, Baseler WA, Davies LC, Ghesquière B, Maio N, Rice CM, Rouault TA, Cassel T., Higashi RM, Lane AN, Fan TWM, Wink DA, McVicar DW. 2020. Оксид азота управляет метаболической перестройкой в макрофагах M1, воздействуя на аконитазу 2 и пируватдегидрогеназу. Связь с природой 11: 1–17. DOI: 10.1038 / s41467-020-14433-7.
Таблица 1 требует дополнительных ссылок.
Спасибо за этот комментарий. В таблице 1 мы добавили дополнительную информацию о фенотипах макрофагов и увеличили количество ссылок.
На мой взгляд, обзорная рукопись требует более одной ссылки для многих моментов, затронутых в рукописи.
Спасибо за эту заметку. Мы добавили больше ссылок, чтобы поддержать высказанные мысли в Обзоре (отмечены желтым).
Абзацы заканчиваются и начинаются, не формируя целостной мысли.
Спасибо за записку. По вашей рекомендации мы добавили вводную часть и заключение к параграфам, относящимся к неклиническим разделам рукописи . (строки 186 - 190, 219 - 220, 256 - 261, 263 - 266, 355 - 359, 400 - 401, 453 - 458, 460 - 462, 504 - 510, 537 - 557).
Таблицу 2 трудно читать, но я полагаю, что ее можно улучшить путем переформатирования.
Спасибо за ваш ценный комментарий. Мы изменили Таблицу 2, чтобы упростить понимание.
Заключение может быть расширено, чтобы объединить обзор в рассказ. В обзоре содержится много информации, но можно было бы использовать дополнительные комментарии о будущем направлении исследований, касающихся роли макрофагов в эндометриозе и раке, и о том, как связанные исследования могут быть применены к потенциальным методам лечения.
Спасибо, мы расширили раздел «Заключение» и надеемся, что он кажется логическим завершением обзора в его нынешнем виде.
Reviewer 2 Report
This is a well-written manuscript describing comprehensive information about comparative analysis of macrophage behaviors in endometriosis and cancer. The strength of the manuscript is that it highlighted the information from past and recent literature on experimental modulation of macrophage phenotypes in preclinical models of endometriosis and cancer. Also, it describes how both phenotypes of macrophages (M1 and M2) plays in important role in both endometriosis and cancer. The another strength of this manuscript is that it highlighted some recent therapeutics approaches which used macrophages or attractants which induces macrophage polarization along with some limitations associated with those studies. This will help understanding current limitations of these approaches and further drive the research efforts in this area in deciphering the role of macrophages in this disease. The only weakness of this study is that there are not enough references provided for some information and previous literature throughout the manuscript. This will need to be addressed before the manuscript can be published. In conclusion, the review is well-described overall and the content are correct and update.
Author Response
We thank the reviewer for carefully reading the manuscript, positive feedback and interest in our research, fair comments and criticism. Added links to confirm the materials described in the review (marked in yellow) both in the text and in the tables. See Appendix below.
Мы благодарим рецензента за внимательное чтение рукописи, положительные отзывы и интерес к нашему исследованию, справедливые комментарии и критику. Добавлены ссылки для подтверждения материалов, описанных в обзоре (отмечены желтым) как в тексте, так и в таблицах. См. Приложение ниже.
Round 2
Reviewer 1 Report
I am satisfied with these revisions. The manuscript reads great now and the references are adequate.
Author Response
Thank you!